# Is there an indication for simultaneous bilateral knee arthroplasty in morbidly obese patients? Should the patients' request for simultaneous operation be considered?

**Bedrettin Akar** [ID] *

Deparmant of Orthopedics and Traumatology, Sakarya Yenikent State Hospital, Sakarya, Turkey

* drbedrettin@gmail.com

## Abstract

### Purpose

This study aimed to analyse the safety of simultaneous bilateral total knee arthroplasty (SBTKA) surgery by comparing morbidly obese (MO) patients with obese patients.

### Methods

SBTKA was performed to 494 patients by a single surgeon in a single center between 2014–2020. The patients followed for a mean of 26 months. They were divided into two groups according to body mass index (BMI) as MO (BMI> 40 kg/m2, n = 65) and obese(O) (BMI = 30–39.9 kg/m2, n = 429 patients). The groups were compared in terms of wound healing problems (WHP), mobilization time (MT), operation time (OT), prosthesis infection, aseptic loosening (AL), early complications, revision, and length of hospitalization using univariate and multivariate logistic regression analyses.

### Results

Logistic regression analysis revealed significant differences in the clinical outcomes and complications between MO and O patients. Parameters such as length of stay, OT, MT, WHP, debridement, medial retinaculum detachment (MRD) and AL, and short-term complications such as acute kidney injury (AKI), and pulmonary embolism (PE) incidence were significantly higher in the MO group. Among the MO patients, the clinical outcomes were worse than those among the O patients, and the complication incidence was higher.

### Conclusion

We do not find SBTKA surgery feasible in morbidly obese patients due to the high complication rate and unsatisfactory clinical outcomes. We suggest that the patient's request to undergo SBTKA should not be taken into account, and that staged surgery be preferred.

**Data Availability Statement:** All relevant data are within the paper.

**Funding:** The author(s) received no specific funding for this work.

**Competing interests:** The authors have declared that no competing interests exist.

## Introduction

Obesity has become a universal health problem and its prevalence is increasing daily. Considering that 40% of the world population is obese, this clinical problem continues with increasing incidence [1]. Obesity is a preventable-modifiable risk factor. Studies examining the trends in obesity have shown that its prevalence has raised in all age groups, regardless the ethnicity or socioeconomic status [1, 2]. From 1999 through 2018, the obesity prevalence in the United States increased from 30.5% to 42.4%. Notably, the obesity prevalence was 40% among adults aged 20 to 39 years, 45% among adults aged 40 to 59 years, and 43% among the group aged 60 years and older. Obesity is a strong risk factor for osteoarthritis of the knees [3]. The incidence of total knee arthroplasty (TKA) is high in obese patients. The frequency of simultaneous bilateral TKA varies in the literature but is always low, between 2 and 7% of all TKA. Although the exact mechanism is not understood, it is predicted that excessive joint loading changes the walking and movement strategies in O patients, which in turn causes deterioration of the alignment and cartilage degeneration in the joints. In addition, obesity-associated dyslipidemia, proinflammatory adipokines and cytokines are thought to induce joint damage [3, 4].

The probability of total knee arthroplasty (TKA) due to arthrosis is high in MO patients. Usually, symptoms and degeneration are bilateral, and TKA is indicated in both knees. Whether SBTKA is the right decision in MO patients is highly controversial. In these patients, surgery is both technically more difficult and longer [4]. Morbid obesity is a complex disease associated with a number of comorbidities such as cardiac diseases (CD), diabetes mellitus (DM), sleep apnea (SA), respiratory problems, and hypertension (HT). Superficial fat necrosis and WHP are common in MO patients [5]. Due to the negative effects of obesity on the immune system and decreased subcutaneous tissue oxygenation, the rate of infection development in these patients is quite high, which increases the probability of a prosthesis infection [5, 6]. It should not be forgotten that SBTKA has advantages such as high patient satisfaction, short general rehabilitation period and low cost.

In MO patients, SBTKA has various disadvantages, such as long hospitalization, mobilization, and OT, WHP, infection, AL, slow functional rehabilitation, and rupture of bilateral deep incisional sutures due to falling.

Our retrospective study aimed to analyze the safety of SBTKA with univariate and multivariate logistic regression analyzes in MO patients

## Materials and methods

SBTKA was performed to 494 patients by a single surgeon in a single center between January 2014- February 2020.Ethics committee approval was obtained from Sakarya University Faculty of Medicine on 03.01.2022 with document number 92632-545.In our retrospective study, the ethics committee waived the requirement for informed consent. All patients with severe knee arthrosis (Grade IV) were included in the study. The patients were followed up for a mean of 26 (12–45) months and divided into MO (BMI> 40 kg/m2), and O (BMI = 30–39.9 kg/m2) groups based on their BMI values. The MO group had 65 patients, with a mean age of 67.51 ±7.28 years, while the O group had 429 patients with a mean age of 71.59±7.04 years. The weight of all patients was measured by the professionals in our clinic the day before the operation. Mean BMI index was 44.8 kg/m2 in MO patients and 35.2 kg/m2 in O patients. BMI index was 40–45 kg/m2 in 58 of the MO patients and 45–50 kg/m2 in the remaining 7 patients. All of the MO patients had comorbidities such as HT, DM, cardiac and respiratory problems, vascular and renal disorder. Patients who underwent staged SBTKA, non-obese patients, and

those receiving oncological treatment were excluded from the study. TKA was performed to the patients under tourniquet and epidural anesthesia, first on the right and then on the left. Tranexamic acid was not used in the patients. The surgical technique was performed using a midline incision and median parapatellar deep exposure. The patella was translated laterally with eversion. Extramedullary tibial alignment was accomplished with a plan to resect 3 mm from the less affected side. Intramedullary femoral alignment was performed with a gap balancing approach. Patellar resurfacing was routinely accomplished. Wound closure was accomplished with vicryl sutures for the retinacular repair, and staples for the skin closure. Cemented total knee prostheses of different brands (Wright, Stryker, Biomed, Concensus) protecting the posterior cruciate ligament were used. The patients were started on antithrombotic stockings together with prophylactic antibiotics (cephazolin 3x1 g), and an anticoagulant (enoxaparin 1x0.4–0.8ml) one day before the operation. Anticoagulant prophylaxis was administered up to four weeks post op. Anticoagulant prophylaxis was applied at higher doses in the MO group than in the O group. Prophylactic antibiotics were administered at the same doses in MO and O patients. Antibiotic doses were not increased in morbidly O patients in order not to adversely affect renal functions. No additional antibiotic dose was needed before the second operation. The patients were followed up in the 2nd week (to assess early wound complications), 6th week, 3rd month, first year, and annually thereafter. All groups were compared in terms of WHP, MT, OT, AL, prosthesis infection, early complications (deep vein thrombosis, AKI, PE), revision, and the length of hospitalization using univariate and multivariate logistic regression analyses. All outcomes in the manuscript are surgical complications and are not related to cost analysis, quality of life, patient-reported outcomes.

In the power analysis conducted with the G*power 3.1 program, the effect size of the study groups for the MT was found to be 0.46 (The outcomes of TKA in MO patients: a systematic review of the literature) (alpha error probability = 0.05); In the sample size analysis performed with a power value of 0.80, the total number of samples required to be taken was found to be 120 (at least 60 for each group).

## Statistical evaluation

In this study, statistical analyses were performed with NCSS (Number Cruncher Statistical System) 2007 Statistical Software (Utah, USA) package program (Table 1).

In addition to descriptive statistical methods (mean, standard deviation), the Shapiro Wilk test was used to assess the normality of data distribution. The independent t-test and the Mann Whitney U tests were used for the comparison of normally and non-normally distributed variables between paired groups. The chi-square and Fisher's reality tests were used for comparing qualitative data. To determine the factors affected by MO, multivariate logistic regression analyses were performed with the variables significant in the univariate tests. The results were evaluated at the significance level of $p < 0.05$.

## Results

Multivariate and univariate logistic regression analyses of both groups revealed significant differences in clinical outcomes and complications (Table 2). The initial mobilization period of the patients ranged between 1–3 days, and were 1.98±0.63 days among MO patients, and 1.41 ±0.57 among the O group. The operation times of the MO and O groups were 89.77±14.7 minutes, and 82.18±14.68 minutes, respectively. The MO patients were hospitalized for 9.62±1.68 days, and the O patients, for 7.31±2 days (Fig 1). Among 494 patients who underwent SBTKA, superficial wound healing problems accompanied by serous discharge developed in 14 (21.54%) patients in the MO group and 21(4.90%) patients in the O group(p = 0,0001).

**Table 1. Statistical analysis.**

| | | Obese Group n:429 | | Morbidly obese Group n:65 | | p |
|---|---|---|---|---|---|---|
| Age | | 71,59±7,04 | | 67,51±7,28 | | **0,0001*** |
| Gender | Male | 36 | 8,39% | 3 | 4,62% | 0,293+ |
| | Female | 393 | 91,61% | 62 | 95,38% | |
| Hospitalization Time (Days) | | 7,31±2 | | 9,62±1,68 | | **0,0001*** |
| Operation Time (Minutes) | | 82,18±14,68 | | 89,77±14,7 | | **0,0001*** |
| Mobilization Time (Days) | | 1,41±0,57 | | 1,98±0,63 | | **0,0001*** |
| Wound healing problems | None | 408 | 95,10% | 51 | 78,46% | **0,0001+** |
| | Present | 21 | 4,90% | 14 | 21,54% | |
| Debridement | No | 427 | 99,53% | 61 | 93,85% | **0,003‡** |
| | Yes | 2 | 0,47% | 4 | 6,15% | |
| Medial Retinaculum Separation | None | 422 | 98,37% | 56 | 86,15% | **0,0001+** |
| | Present | 7 | 1,63% | 9 | 13,85% | |
| Long–term complications | None | 414 | 96,50% | 58 | 89,23% | - |
| | Aseptic Loosening | 7 | 1,63% | 4 | 6,15% | **0,018‡** |
| | Septic Loosening | 2 | 0,47% | 1 | 1,54% | 0,333‡ |
| | Perprostatic Femur Fracture | 6 | 1,40% | 2 | 3,08% | 0,603‡ |
| Short–term complications | None | 395 | 92,07% | 51 | 78,46% | - |
| | AKI | 24 | 5,59% | 10 | 15,38% | **0,005+** |
| | DVT | 9 | 2,10% | 2 | 3,08% | 0,831‡ |
| | Pulmonary Embolism | 1 | 0,23% | 2 | 3,08% | **0,038‡** |
| Diabetes Mellitus | | 68 | 15,85% | 13 | 20,00% | 0,400+ |
| Hypertension | | 216 | 50,35% | 36 | 55,38% | 0,449+ |
| Ischemic Heart Disease | | 16 | 3,73% | 8 | 12,31% | **0,003+** |
| Renal Disease | | 16 | 3,73% | 5 | 7,69% | 0,140+ |
| Pulmonary Disease | | 10 | 2,33% | 4 | 6,15% | 0,083+ |
| Venous insufficiency- lymphedema | | 7 | 1,63% | 3 | 4,62% | 0,111+ |
| Thyroid Disease | | 14 | 3,26% | 2 | 3,08% | 0,937+ |

*Independent t test

+ Chi-square test

‡Fisher's Reality Test

Debridement was required in 4(6,15%)and 2(0.47%)patients in the MO and O groups, respectively, due to the spread of the infection to the subcutaneous tissues (p = 0,003).There were no adverse effects on the implants in the debrided cases. The patients recovered without sequelae with regular wound care and antibiotic therapy. Mild lymphedema was observed in 3(4.62%) patients in the MO group and in 7(1.63%) patients in the O group (p = 0,111). No treatment planned for lymphedema. Prosthesis infection was detected in 1(1.54%) patient in the MO group and 2(0.47%) patients in the O group (p = 0,333). After the diagnosis of prosthesis infection, specific antibiotherapy was started according to the antibiogram results (Fig 2). The infections did not respond to antibiotherapy, the prosthesis was removed, and a spacer with antibiotics was placed with plans of future revision knee arthroplasty. Periprosthetic supracondylar femur fracture developed due to fall in 2(3.08%) MO patient and 6(1.40%) O patients (p = 0,637). Internal fixation was performed with a plate and screw for these fractures. Soft tissue repair surgery was performed in 9(13.85%) of the MO patients and in 7(1.63%) of the O patients during the early postoperative period, since extensor mechanism insufficiency developed due to the rupture of deep surgical sutures due to falling during mobilization

**Table 2. Logistic regression analysis.**

| | Univariate Risk | | Multivariate Risk | |
|---|---|---|---|---|
| | OR (95% CI) | p | OR (95% CI) | p |
| **Age** | 0.92 (0.88–0.96) | **0.0001** | 0.92 (0.87–0.97) | **0.002** |
| **Hospitalization Time (Days)** | 1.64 (1.43–1.88) | **0.0001** | 1.74 (1.46–2.07) | **0.0001** |
| **Duration of surgery (Hour)** | 1.02 (1.01–1.05) | **0.0001** | 1.03 (1.00–1.05) | **0.009** |
| **Mobilization time (days)** | 4.13 (2.67–6.35) | **0.0001** | 3.36 (2.01–5.61) | **0.0001** |
| **Medial Retinaculum Detachment** | 4.68 (2.47–7.04) | **0.0001** | 1.53 (1.10–4.31) | **0.002** |
| **Aseptic Loosening** | 0.25 (0.07–0.87) | **0.029** | 1.02 (0.20–4.09) | 0.806 |
| **Wound Healing Problems** | 5.33 (2.55–8.14) | **0.0001** | 0.76 (0.55–1.57) | **0.005** |
| **Debridement** | 4.01 (2.51–8.06) | **0.003** | 0.13 (0.01–3.03) | 0.201 |
| **AKI** | 3.23 (1.46–7.13) | **0.004** | 0.76 (0.23–1.48) | 0.642 |
| **Pulmonary Embolism** | 5.49 (1.38–9.87) | **0.026** | 0.94 (0.06–1.85) | 0.961 |
| **Ischemic heart disease** | 3,62 (1,48–8,85) | **0,005** | 0,21 (0,05–0,78) | 0,021 |

(p = 0,0001). Soft tissues were repaired by surgery for the third time. AL was observed in 4 (6.15%) patients in the MO and 7(1.63%) patients in the O groups. It was diagnosed by bone scintigraphy when the patients felt severe pain during weight bearing(p = 0,064). These patients underwent revision surgery in the early period. While DVT, AKI, and PE developed in 2(3.08%), 10(15.38%), and 2(3.08%) patients, respectively, in the MO group, it was observed in 9(2.10%), 24(5.59%), and 1(0.23%) patient, respectively, in the O group. While there was a statistically significant difference in terms of AKI(p = 0,05) and PE(p = 0,038), there was no significant difference in terms of DVT(p = 0,831). All complications healed uneventfully with conservative treatments. Joint instability did not develop in MO patients due to the use of cruciate retaining (CR) implants. In terms of stability, We think that cruciate retaining implants can be used safely in MO patients. The clinical results of the MO patients were worse than those of the O patients, and the complication incidence was higher.

## Discussion

According to the criteria of the World Health Organization, those with a BMI >30kg/m2 are considered obese, and those with a BMI >40kg/m2 are considered morbidly obese. While SBTKA surgery is still a matter of debate among orthopedists, there are serious concerns about performing SBTKA to O and MO patients. Obesity is a strong risk factor for bilateral osteoarthritis of the knees [6–8]. Close to 40% of the world population and 900 million people worldwide are obese, and the numbers increase daily. MO and O patients are mostly female. Sixty-two of the MO patients in our study were female and 3 were male. Obesity has been suggested as a modifiable risk factor [8, 9]. TKA is challenging in various ways in O and MO patients, starting from the associated comorbidities, positioning, long operation time, high risk of intraoperative bleeding, long hospitalization period, delayed wound healing, high infection rates, aseptic implant loosening, rupture of superficial and deep surgical sutures due to early postoperative fall, and medial collateral ligament avulsion. There are difficulties encountered during surgery and in the early postoperative period. Negative events seen in the morbidly obese group are not specific to bilateral knee replacement surgery performed under single anesthesia, but are also encountered in unilateral or staged bilateral knee surgery. However, MO paves the way for the development of negative events in bilateral knee surgery performed with a single anesthesia due to both high weight and accompanying comorbidities. As BMI increases, perioperative morbidity and mortality rates in SBTKA continue to be a source of

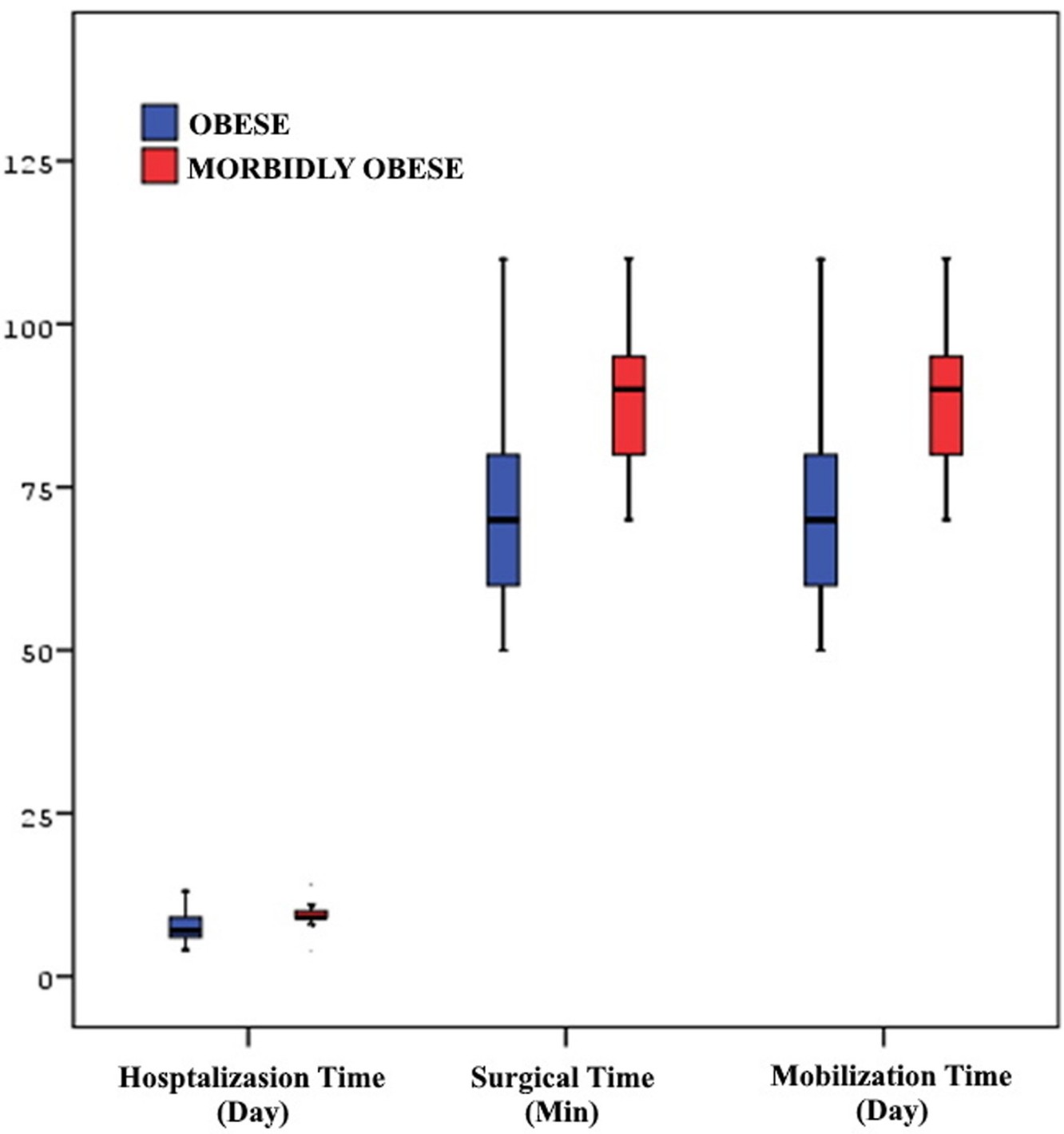

**Fig 1. Obese, morbidly obese, hospitalization time, surgical time, mobilization time.**

concern [9, 10]. Studies showing the effects of morbid obesity on SBTKA are very few in the literature, and research comparing the obese and non-obese patients were performed on one knee. MO is a complex disease associated with a number of comorbidities such as HT, CD, DM, SA, respiratory problems, and stroke [10–12]. Logistic regression analyses of the MO and

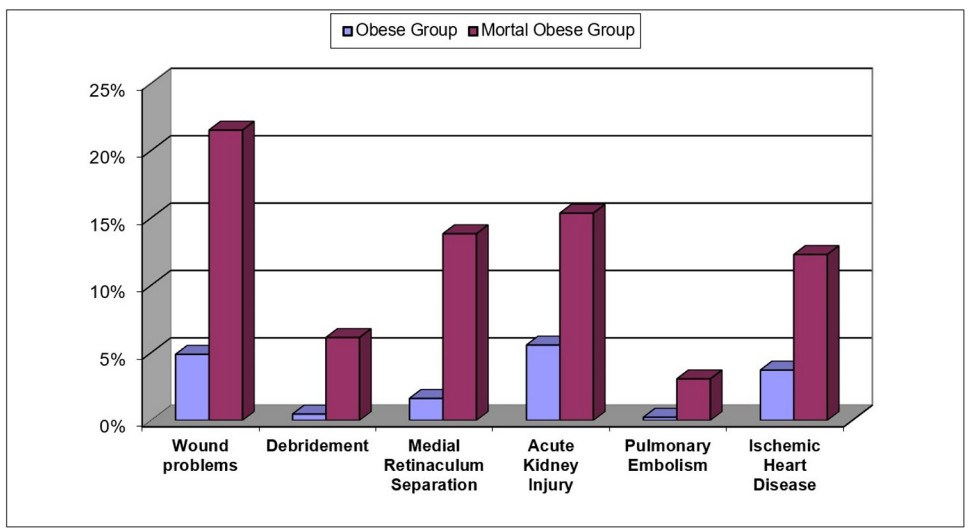

**Fig 2. Parameters showing significant differences between the obese and morbidly obese.**

obese patients showed significant differences in postoperative clinical parameters and complications in our study. We think that significant differences in length of hospital stay, duration of surgery and mobilization negatively affect the clinical outcome and trigger complications. Complications such as WHP, MRD, and aseptic and septic loosening are more common in MO patients compared to obese patients. Adverse events in MO patients should not be attributed to SBTKA. However, it should not be forgotten that MO can pave the way for these adverse events. In addition, many comorbidities of MO patients will increase the risks and adverse events due to the stress caused by bilateral surgery.

In their prospective study on 529 TKAs, Dowsey et al. reported the incidence of adverse effects as 35.1% in MO patients, 22.1% in the obese and 14.2% in the non-obese. They stated that there were significant differences between the groups, and an increase in complications was seen at a rate of 58% among the O population [13]. McElroy et al. found the complication rates as 22% in the MO group, 15% in the O group and 9% in the non-obese group [14]. In our study, we found that early and late complications (AKI, PE, AL, and wound problems) were higher among the MO patients who underwent SBTKA compared to obese patients. In our study, in accordance with the literature, the complication rate was found to be 34.4% in MO patients and 11.4% in O patients.

Ogur et al. found that unilateral and SBTKA procedures did not differ in clinical scores, or perioperative and postoperative complications among the non-obese and O patients. However, complications were moderately increased in MO patients who underwent unilateral TKA, and significantly increased among the MO who underwent SBTKA [15]. Al Turki et al. stated that BMI was not associated with long surgical duration in both unilateral and SBTKA; therefore, BMI should not be considered an indicator for long-term operation when performing TKA [16]. However, in many studies and ours, MO prolonged the already long OT (since dissection of the tissues is difficult) of SBTKA and paved the way for the various complications.

DeMik et al. emphasized that MO has a significant effect on postoperative complications, and it is important to consider patient comorbidities when evaluating surgical risks [17]. In our study, general and systemic complications such as DVT, AKI, and PE were observed. Remily et al. reported that orthopedists should review the risks and benefits of SBTKA in detail,

since the overall complication rates are high in both O and MO patients and will cause a long hospital stay [18].

Goldstein et al. stated that revision surgeries were common among the MO and O patients who underwent SBTKA due to disruption of the extensor mechanism as a result of patellar tendon and quadriceps tendon rupture. Soft tissue repair was performed for the third time, as 3 of 9 MO patients fell again and re-ruptured the extensor mechanism [19].

Boyce et al. reported that perioperative complications such as revision rates and superficial wound infection are high in MO patients [20]. In our study, superficial WHP developed in 14 MO and 21 O patients. The wounds later healed uneventfully with wound care treatment. Sezgin et al., and Giesinger et al. associated MO with infection and increased risk of revision. They stated that the risks of infection and AL in SBTKA surgery increase with BMI [21, 22]. In our study, the incidence of AL in MO was higher compared to the obese group.

While it is stated that SBTKA can be performed successfully and safely in patients with BMI <40kg/m2, it is considered as a very risky and low-success-rate intervention, especially in patients with BMI>40kg/m2.

Our study had some limitations. One of the limitations of our study was that the surgical population was predominantly female, therefore the results of the study could not be valid for male patients. In addition, the retrospective nature of the study and the assessment of only obese patients were also limitations of our study. Furthermore, the fact that all patients were operated by the same surgeon and in a single center was another limitation of the study.

## Conclusion

SBTKA is already a major surgical procedure with its own risks and complications. When MO and accompanying comorbidities are added, it is obvious that complications will increase, and our surgical success rate will decrease. Although we did not encounter mortality in MO patients who underwent SBTKA in our study, we do not recommend performing SBTKA in MO patients due to increased comorbidities and perioperative and postoperative risks. We suggest that the request of MO patients to have SBTKA should not be taken into account by the orthopedists, and that patients should be encouraged to lose weight first. If they cannot lose weight and insist on having surgery, staged surgery should be performed, considering the comorbidities of these patients.

We hope that this study will shed light on the treatment of morbidly obese patients, which is an increasing problem of our age, and we think that studies involving more patients and more parameters are needed.

## Author Contributions

**Conceptualization:** Bedrettin Akar.

**Data curation:** Bedrettin Akar.

**Formal analysis:** Bedrettin Akar.

**Funding acquisition:** Bedrettin Akar.

**Investigation:** Bedrettin Akar.

**Methodology:** Bedrettin Akar.

**Visualization:** Bedrettin Akar.

**Writing – original draft:** Bedrettin Akar.

**Writing – review & editing:** Bedrettin Akar.

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
