## [Decision Letter · Decision Letter 0]

20 Apr 2023

PONE-D-22-20821IS THERE AN INDICATION FOR SIMULTANEOUS BILATERAL KNEE ARTHROPLASTY IN MORBIDLY OBESE PATIENTS? SHOULD THE PATIENTS’ REQUEST FOR SIMULTANEOUS OPERATION BE CONSIDERED?PLOS ONE

Dear Dr. Akar,

Thank you for submitting your manuscript to PLOS ONE. After careful consideration, we feel that it has merit but does not fully meet PLOS ONE’s publication criteria as it currently stands. Therefore, we invite you to submit a revised version of the manuscript that addresses the points raised during the review process.

We look forward to receiving your revised manuscript.

Kind regards,

Sameh Attia, MS

Academic Editor

PLOS ONE

Journal Requirements:

Reviewers' comments:

Reviewer's Responses to Questions

**Comments to the Author**

1. Is the manuscript technically sound, and do the data support the conclusions?

Reviewer #1: Partly

Reviewer #2: Yes

Reviewer #3: Yes

2. Has the statistical analysis been performed appropriately and rigorously? 

Reviewer #1: Yes

Reviewer #2: Yes

Reviewer #3: Yes

3. Have the authors made all data underlying the findings in their manuscript fully available?

Reviewer #1: Yes

Reviewer #2: Yes

Reviewer #3: Yes

4. Is the manuscript presented in an intelligible fashion and written in standard English?

Reviewer #1: No

Reviewer #2: Yes

Reviewer #3: Yes

5. Review Comments to the Author

Reviewer #1: Thank you for the submission of your manuscript to PLOS ONE.

The consideration for total knee arthroplasty in morbidly obese patients---especially single-anesthetic (simultaneous) bilateral TKA----is becoming an increased point of controversy. As the economics of healthcare delivery are shifting, including the translation of care for surgical complications on treatment centers, it is relevant to understand what the overall risks of complications are for this unique cohort. The study does this well.

The main critique that I anticipate readers will have for this study is whether this fully answers the question about safe bilateral knee replacement. While the authors have demonstrated substantially lower complication rates among obese patients than morbidly obese patients, the question will arise whether the authors are able to provide any insight into a BMI threshold that might exist for "safe" single anesthetic bilateral TKA----when considering morbid obesity as an independent risk factor. The manuscript does not provide this information.

There are considerations about the generalizability of this study to the collective body of patients who might undergo a single-anesthetic bilateral total knee replacement surgery. In this surgical population, it appears that the majority of patients (obese and morbidly obese) were female. This study cannot adequately assert whether the findings of this study will equally apply to male patients. This should be specifically stated among the study limitations.

The manuscript is generally well written with minimal syntax/punctuation concerns, that should be able to be addressed in the process of final manuscript editing.

Specific Considerations Listed Below, along with the points in the manuscript where they occur:

1) Materials and Methods section (Line 13 on page): “respiratory problems, vasculer and renal disorder.” Correct spelling is “vascular”

2) Materials and Methods section (8 lines from the bottom of page 4); “ Surgical technique was as follows: Appropriate femoral and tibial osteotomies were performed by tilting the patella laterally with an anterior long incision. Cemented total knee prostheses of different brands (Wright, Stryker, Biomed, Concensus) protecting the posterior cruciate ligament were used. After the tourniquet was loosened, bleeding was controlled, an aspirative drain was placed and the layers were closed anatomically.

-----Suggesting this should be rewritten with the following lay-out: The surgical technique was performed using a midline incision and (median parapatellar, mid-vastus, or sub-vastus) deep exposure. The patella was translated laterally (with/without) eversion. Intramedullary/Extramedullary tibial alignment was accomplished with a plan to resect (XX mm from the more affected/less affected side). Intramedullary/Extramedullary femoral alignment was performed with a (measured resection/gap balancing) approach. Patellar resurfacing (was/was not) routinely accomplished. Wound closure was accomplished with (XXX sutures for the retinacular repair, and staples/sutures for the skin closure).......Then continue with postoperative management.

3) Materials and Methods (last line on Page 4): “Anticoagulant prophylaxis was administered up to four weeks postop.” Was enoxaparin used during the postoperative course, or were other agents used. The note indicates that the duration of anti-coagulant medication was “up to four weeks postop.” Were there patients receiving anticoagulation for a different time interval than 4 weeks. If yes, what would determine how long prophylaxis was used? What proportion of patients received 2 weeks or less? Did the duration of prophylaxis affect venous thromboembolism (VTE) events?

4) Table 1- Morbidly obese patients undergoing single-anesthetic bilateral TKA in this institution are predominantly female. This constituted 91.6% of the obese patient group and 95.4% in the morbidly obese group. This finding may limit the generalizability of the study findings to a limited (female) patient population. (previously noted above: This should be noted in the study limitations on generalizability).

5) Table 1- There was a slightly longer time to ambulation in the morbidly obese group. Can the authors discuss how procedures are performed at their institution and any effect that this might have on patient mobility? (ie. Are surgeries performed in a single operating room or do the surgeon(s) use dual rooms or other construct to perform surgical procedures? Are these cases performed early in the day, or are they the last procedure performed in a given day----and does this influence the patients’ time to ambulation? How often are blood transfusions being given to patients? Does this influence the time to ambulation in patients, or is the delay related to other (i.e. physical performance/pain control, etc) considerations? If these cannot be determined, they should be included in the study limitations.

5) Results- First paragraph. Findings (wound healing problems, debridements, etc) should be reported with percentages and p-values (rather than just the numbers).

6) Results- Can the authors report on the timing of the periprosthetic fractures that occurred? Were any of these intraoperative fracture? How many were identified during the initial hospital stay? How many occurred during the first 3 months after surgery? How many occurred during the first year after surgery? How many occurred remote to the surgery (falls unrelated to the procedure/early recovery).

6) Discussion (page 9)- The authors report 40% of the world population (900 million) patients being obese----somewhat relevant since all of the patients in this study were obese. However, since the identified differences relate to morbid obesity, the discussion should also provide a description of the scope of concern associated with morbid obesity, in addition to much larger population with obesity (BMI > 30 kg/m2).

7) Discussion (page 9, 2nd half)- The authors acknowledge the challenges associated with both bilateral TKA and unilateral TKA in the morbidly obese population. “Negative events seen in the morbidly obese group are not specific to bilateral knee replacement surgery performed under single anesthesia, but are also encountered in unilateral or staged bilateral knee surgery.”

- The question(s) arising from this discussion/presentation is whether TKA is a safe operation in patients who are morbidly obese. When comparing SBTKA in the morbidly obese population to SBTKA in the uncomplicated obese population, there are certainly increase complication risks. Can the authors relate how their complication rates in the SBTKA morbidly obese group compares with other studies and what they have reported with either unilateral or bilateral TKA in this patient population? Are the authors’ rates of complications similar to what others have reported? Are the rates of complications in SBTKA greater than what they are in staged TKA? A consideration that SBTKA should be avoided might be followed by a question whether primary TKA should be avoided/delayed for this patient cohort…..How would the authors answer this question? Noted that in the Discussion (Page 10)- The authors present two studies reporting on the rates of complications in non-obese, obese, and morbidly obese patients. The rates of complications/adverse events were 35.1% and 22% in the two studies for the morbidly obese population. The authors should be able to make a statement in this part of the discussion about how their complication rates compared relative to these other studies (better or worse?)

8) Discussion page 10 (3-5 lines from the bottom): “However, in many studies and ours, MO prolonged the already long OT (since dissection of the tissues is difficult) of SBTKA and paved the way for the various complications.”

- How strongly is this statement supported by the study? True, there was a statistically significant increase in the operative time between the two cohorts. But the mean difference in OR time was only 7 minutes. It does not seem that this is really long enough to result in the broad differences in complications that were reported between the two cohorts. Were the cases associated with complications associated with a prolonged operative time? Did they only occur in the most extreme operative time cases, or were they distributed across the entire range? Unless the operative time was close to or exceeding 120 minutes in those cases, it might be difficult to conclude that the “already long OT….paved the way for the various complication.”

9) Discussion of Limitations (page 11, last paragraph): “Our study had some limitations, one of the most significant of which was the scarcity of other research on MO in SBTKA.” This isn’t a limitation of this study. It may be a limitation of the literature on the subject---which is a good reason why the study is being performed. Agree that the current weaknesses include the assessment of only obese patients and the retrospective nature of the study. Because patients in both cohorts had other risk factors this may make it more difficult to understand whether some complications are related to the severity of obesity or the other concurrent conditions (?).

10) Table 2- The LRA is presenting the different outcomes, with focused consideration of morbid obesity vs obesity. Can the authors define what concurrent conditions were also configured into the model? The reviewer is expecting to see information evaluated with multi-variate instruments assessing the relative contributions of different comorbidities to risk---particularly when individuals may have multiple concurrent risk factors (obesity, diabetes, heart disease, pulmonary disease/sleep apnea, etc). In addition to the post-surgical outcomes present in this table, can the authors also provide an assessment of the different risk factors and whether morbid obesity status had a more significant independent impact on the outcome variables than other co-morbidities?

This is a potentially valuable work, although the reviewer is not certain how many surgeons (internationally) perform single anesthetic bilateral TKA on patients who are morbidly obese. The authors have certainly demonstrated how this results in a substantial number of patients who experience complications (compared with non-morbidly obese patients). But, how will this study impact practice, if most surgeons don't perform SBTKA in this patient population?

The reviewer still wonders if the authors could provide an "upper limit" for safe SBTKA. Is it really 39.9 kg/m2?

Thank you for your interest in this subject matter, for reviewing your experience, and for sharing this information. Best wishes as you continue to refine your work.

Reviewer #2: In this article, Bedrettin Akar presents the results of his comparative study in patients with simultaneous bilateral TKA and obesity. The patient groups differ primarily in their BMI, below and above a BMI value of 40 kg/m2. The author was able to include a total of 494 patients in the study, with a large difference in group size (65 vs. 429 patients).

Statistical analysis of the retrospective data was performed about wound healing problems (WHP), mobilization time (MT), operation time (OT), prosthesis infections, aseptic loosening (AL), early complications, revisions, and length of hospital stay. As expected, patients in the group with BMI >40 showed significantly more complications and worse clinical outcomes of arthroplasty. Therefore, the author recommends that patients with BMI >40 kg/m2 should be encouraged to reduce their weight. If this is not possible, SBKA should not be performed and the patient should be aggressively educated about the limitations, complications, and dangers.

The article contributes to the care of obese patients, whose numbers are steadily increasing and pose a increasing challenge to healthcare systems.

Therefore, we consider the article worthy of publication.

However, we again recommend a critical review. For example, the author should explain abbreviations when they are first used (see Introduction: MO). In addition, the differences between the groups would be even clearer if a higher degree of matching had been done preoperatively (age, sex). Here, Scandinavian studies on peri-implant issues show significant differences, but these studies are based on the analysis of large register cohorts, which were not available to the author. Perhaps he can briefly explain here why, in his opinion, it was not possible matching the groups even better.

Reviewer #3: Bedrettin et al, aimed to analyze the safety of simultaneous bilateral total knee arthroplasty (SBTKA) surgery in morbidly obese patients compared to obese patients. The study found that SBTKA surgery is not feasible in morbidly obese patients due to the high complication rate and unsatisfactory clinical outcomes. The authors suggest that staged surgery be preferred and that the patient's request to undergo SBTKA should not be taken into account. The study used appropriate statistical analysis methodology, including univariate and multivariate logistic regression analyses, descriptive statistical methods, and tests for normality and comparison of variables between groups. However, the study has some limitations, such as the retrospective nature of the study which was addressed shortly by the authors and the use of one surgeon for all cases which has to be discussed as a limitation. The main findings of the study were adequately discussed in the Results and Discussion sections of the paper. Overall, the study provides valuable insights into the safety of SBTKA surgery in morbidly obese patients and highlights the need for further research on the treatment of morbidly obese patients.

Minor issues are the required a detailed check to enhance its readability, consistent writing styles between British and American English has to be corrected (ex. Analyzes, analyses). Also it is confusing have one group abbreviated (MO and the other with full name) I recommend using MO group and O group throughout the manuscript.

However, two major issues shall be addressed:

1- The introduction: The authors briefly mention the high prevalence of obesity and the increased risk of complications associated with TKA surgery in obese patients. However, the introduction does not provide a detailed discussion of the critical need for this study in the field. The authors do not provide a comprehensive review of the existing literature on the safety of simultaneous bilateral TKA surgery in morbidly obese patients, nor do they discuss the limitations of previous studies. Therefore, while the introduction provides some motivation for the study, it should have been more comprehensive in its discussion of the critical need (gap of knowledge) for this study in the field. Furthermore, the frequency of bilateral operation in these cases should be introduced locally and internationally.

2- the fact that all the patients were operated by the same surgeon and in one center is a limitation of the study. This is because the results of the study may not be generalizable to other centers or surgeons, as different surgeons may have varying levels of experience and skill, and different centers may have different resources and protocols for managing surgical procedures. Additionally, having a single surgeon and center involved in the study may introduce bias into the results. For example, the surgeon may have a preferred technique or approach that could influence the outcomes of the surgery, and the center may have different patient populations or healthcare resources that could impact the results.

Therefore, it is important to interpret the study's findings within the context of this limitation, and to consider the need for further research in other centers and with different surgeons to confirm the study's results and to better understand the safety and effectiveness of simultaneous bilateral knee arthroplasty in morbidly obese patients.

6. PLOS authors have the option to publish the peer review history of their article (what does this mean?). If published, this will include your full peer review and any attached files.

Reviewer #1: No

Reviewer #2: **Yes: **Chris Biehl

Reviewer #3: No

---

## [Author Response · Author response to Decision Letter 0]

8 May 2023

RESPONSE TO REVIEWERS

Reviewer #1:

The manuscript is generally well written with minimal syntax/punctuation concerns, that should be able to be addressed in the process of final manuscript editing.

Specific Considerations Listed Below, along with the points in the manuscript where they occur:

1) Materials and Methods section (Line 13 on page): “respiratory problems, vasculer and renal disorder.” Correct spelling is “vascular”

RESPONSE: The typo was corrected.

2) Materials and Methods section (8 lines from the bottom of page 4); “ Surgical technique was as follows: Appropriate femoral and tibial osteotomies were performed by tilting the patella laterally with an anterior long incision. Cemented total knee prostheses of different brands (Wright, Stryker, Biomed, Concensus) protecting the posterior cruciate ligament were used. After the tourniquet was loosened, bleeding was controlled, an aspirative drain was placed and the layers were closed anatomically.

-----Suggesting this should be rewritten with the following lay-out: The surgical technique was performed using a midline incision and (median parapatellar, mid-vastus, or sub-vastus) deep exposure. The patella was translated laterally (with/without) eversion. Intramedullary/Extramedullary tibial alignment was accomplished with a plan to resect (XX mm from the more affected/less affected side). Intramedullary/Extramedullary femoral alignment was performed with a (measured resection/gap balancing) approach. Patellar resurfacing (was/was not) routinely accomplished. Wound closure was accomplished with (XXX sutures for the retinacular repair, and staples/sutures for the skin closure).......Then continue with postoperative management.

RESPONSE: Thank you for the revision. The appropriate section was rewritten as follows:“The surgical technique was performed using a midline incision and median parapatellar deep exposure. The patella was translated laterally with eversion. Extramedullary tibial alignment was accomplished with a plan to resect 3 mm from the less affected side. Intramedullary femoral alignment was performed with a gap balancing approach. Patellar resurfacing was routinely accomplished. Wound closure was accomplished with vicryl sutures for theretinacular repair, and staples for the skin closure.”

 3) Materials and Methods (last line on Page 4): “Anticoagulant prophylaxis was administered up to four weeks postop.” Was enoxaparin used during the postoperative course, or were other agents used. The note indicates that the duration of anti-coagulant medication was “up to four weeks postop.” Were there patients receiving anticoagulation for a different time interval than 4 weeks. If yes, what would determine how long prophylaxis was used? What proportion of patients received 2 weeks or less? Did the duration of prophylaxis affect venous thromboembolism (VTE) events?

RESPONSE: All patients received only enoxaparin for four weeks in the post-operative period. No other anticoagulant agents were used. The morbidly obese group was administered anticoagulant prophylaxis at higher doses (2x0.4ml) than the obese group (1x0.4 ml).Despite prophylaxis, patients who developed VTE in the post-operative period underwent procedures for embolism treatment.

 4) Table 1- Morbidly obese patients undergoing single-anesthetic bilateral TKA in this institution are predominantly female. This constituted 91.6% of the obese patient group and 95.4% in the morbidly obese group. This finding may limit the generalizability of the study findings to a limited (female) patient population. (previously noted above: This should be noted in the study limitations on generalizability).

RESPONSE: Yes, you are absolutely right. The following statement was added to the manuscript: One of the limitations of our study was that the surgical population was predominantly female, therefore the results of the study could not be valid for male patients.

5) Table 1- There was a slightly longer time to ambulation in the morbidly obese group. Can the authors discuss how procedures are performed at their institution and any effect that this might have on patient mobility? (ie. Are surgeries performed in a single operating room or do the surgeon(s) use dual rooms or other construct to perform surgical procedures? Are these cases performed early in the day, or are they the last procedure performed in a given day----and does this influence the patients’ time to ambulation? How often are blood transfusions being given to patients? Does this influence the time to ambulation in patients, or is the delay related to other (i.e. physical performance/pain control, etc) considerations? If these cannot be determined, they should be included in the study limitations.

RESPONSE: There are four Laminar flow operation rooms for orthopedic surgery in our hospital. Two of these rooms are used for Arthroplasty surgeries. These types of operations are performed as the first case in the early hours of the day. Patients are mobilized on the first day after the operation. Blood transfusions are usually performed in the evening in order not to affect patient mobilization and when values are below Hb:7gr/dl and Htc:20%.

5) Results- First paragraph. Findings (wound healing problems, debridements, etc) should be reported with percentages and p-values (rather than just the numbers).

RESPONSE: All changes have been made and added to the article.

6) Results- Can the authors report on the timing of the periprosthetic fractures that occurred? Were any of these intraoperative fracture? How many were identified during the initial hospital stay? How many occurred during the first 3 months after surgery? How many occurred during the first year after surgery? How many occurred remote to the surgery (falls unrelated to the procedure/early recovery).

RESPONSE: None of the periprosthetic fractures were preoperative or intraoperative. Four of the fractures occurred in the first three months, and five of them occurred in the first year due to falling.

7) Discussion (page 9)- The authors report 40% of the world population (900 million) patients being obese----somewhat relevant since all of the patients in this study were obese. However, since the identified differences relate to morbid obesity, the discussion should also provide a description of the scope of concern associated with morbid obesity, in addition to much larger population with obesity (BMI > 30 kg/m2).

RESPONSE: According to the criteria of the World Health Organization, individuals with a BMI >30kg/m2 are considered obese, and those with a BMI >40kg/m2 are considered morbidly obese.

Additional information has been added to the Discussion section.

8) Discussion (page 9, 2nd half)- The authors acknowledge the challenges associated with both bilateral TKA and unilateral TKA in the morbidly obese population. “Negative events seen in the morbidly obese group are not specific to bilateral knee replacement surgery performed under single anesthesia, but are also encountered in unilateral or staged bilateral knee surgery.”

- The question(s) arising from this discussion/presentation is whether TKA is a safe operation in patients who are morbidly obese. When comparing SBTKA in the morbidly obese population to SBTKA in the uncomplicated obese population, there are certainly increase complication risks. Can the authors relate how their complication rates in the SBTKA morbidly obese group compares with other studies and what they have reported with either unilateral or bilateral TKA in this patient population? Are the authors’ rates of complications similar to what others have reported? Are the rates of complications in SBTKA greater than what they are in staged TKA? A consideration that SBTKA should be avoided might be followed by a question whether primary TKA should be avoided/delayed for this patient cohort…..How would the authors answer this question? Noted that in the Discussion (Page 10)- The authors present two studies reporting on the rates of complications in non-obese, obese, and morbidly obese patients. The rates of complications/adverse events were 35.1% and 22% in the two studies for the morbidly obese population. The authors should be able to make a statement in this part of the discussion about how their complication rates compared relative to these other studies (better or worse?)

RESPONSE: In our study, morbidly obese patients who underwent SBTKA had very high complication rates (34.4%), compared to other studies. Due to the high complication rate in these patient groups, first and foremost, the patient should lose weight, then unilateral or staged surgery should be recommended instead of bilateral surgery. These approaches will contribute to minimizing major complication rates.In our study, in accordance with the literature, the complication rate was found to be 34.4% in MO patients and 11.4% in obese patients.

9) Discussion page 10 (3-5 lines from the bottom): “However, in many studies and ours, MO prolonged the already long OT (since dissection of the tissues is difficult) of SBTKA and paved the way for the various complications.”

- How strongly is this statement supported by the study? True, there was a statistically significant increase in the operative time between the two cohorts. But the mean difference in OR time was only 7 minutes. It does not seem that this is really long enough to result in the broad differences in complications that were reported between the two cohorts. Were the cases associated with complications associated with a prolonged operative time? Did they only occur in the most extreme operative time cases, or were they distributed across the entire range? Unless the operative time was close to or exceeding 120 minutes in those cases, it might be difficult to conclude that the “already long OT….paved the way for the various complication.”

RESPONSE: It is thought that the SBTKA surgical procedure is performed under more difficult conditions in MO patients compared to other patient groups, and this causes an increase in the operation time. Although a small difference of 7 minutes was observed between the two groups in terms of operation time, these 7 minutes and much longer (20-30 minutes) operation times pave the way for complications such as AKI.

10) Discussion of Limitations (page 11, last paragraph): “Our study had some limitations, one of the most significant of which was the scarcity of other research on MO in SBTKA.” This isn’t a limitation of this study. It may be a limitation of the literature on the subject---which is a good reason why the study is being performed. Agree that the current weaknesses include the assessment of only obese patients and the retrospective nature of the study. Because patients in both cohorts had other risk factors this may make it more difficult to understand whether some complications are related to the severity of obesity or the other concurrent conditions (?).

RESPONSE: Yes, you are right. We removed the statement regarding the scarcity of research from the limitations.We acknowledge the retrospective nature of the study and the assessment of only obese patients as limitations of our study

11) Table 2- The LRA is presenting the different outcomes, with focused consideration of morbid obesity vs obesity. Can the authors define what concurrent conditions were also configured into the model? The reviewer is expecting to see information evaluated with multi-variate instruments assessing the relative contributions of different comorbidities to risk---particularly when individuals may have multiple concurrent risk factors (obesity, diabetes, heart disease, pulmonary disease/sleep apnea, etc). In addition to the post-surgical outcomes present in this table, can the authors also provide an assessment of the different risk factors and whether morbid obesity status had a more significant independent impact on the outcome variables than other co-morbidities?. 

RESPONSE: Since the severity of obesity is the main subject of the article, it is the dependent variable of the analysis. According to the independent variables of DM, HT, pulmonary disease, renal disease, venous insufficiency-lymphedema, thyroid disease, there was no statistically significant difference between the groups. When we evaluated the multivariate results according to ischemic heart disease only by adjusting, the results did not change.

Table 2: Logistic Regression Analysis

 Univariate Risk Multivariate Risk Multivariate Risk*

 OR (95% CI) p OR (95% CI) p OR (95% CI) p

Age 0.92 (0.88-0.96) 0.0001 0.92 (0.87-0.97) 0.002 0,93 (0,89-1,18) 0,001

Hospitalization Time (Days) 1.64 (1.43-1.88) 0.0001 1.74 (1.46-2.07) 0.0001 1,73 (1,45-2,05) 0,0001

Duration of surgery (Hour) 1.02 (1.01-1.05) 0.0001 1.03 (1.00-1.05) 0.009 1,01 (0,99-1,08) 0,001

Mobilization time (days) 4.13 (2.67-6.35) 0.0001 3.36 (2.01-5.61) 0.0001 3,01 (1,74-4,87) 0,0001

Medial Retinaculum Detachment 4.68 (2.47-7.04) 0.0001 1.53 (1.10-4.31) 0.002 

0,87 (0,67-2,52) 0,001

Aseptic Loosening 0.25 (0.07-0.87) 0.029 1.02 (0.20-4.09) 0.806 0,54 (0,21-1,32) 0,458

Wound Healing Problems 5.33 (2.55-8.14) 0.0001 0.76 (0.55-1.57) 0.005 2,22 (1,87-4,58) 0,001

Debridement 4.01 (2.51-8.06) 0.003 0.13 (0.01-3.03) 0.201 0,67 (0,16-2,17) 0,684

AKI 3.23 (1.46-7.13) 0.004 0.76 (0.23-1.48) 0.642 0,69 (0,22-1,17) 0,526

Pulmonary Embolism 5.49 (1.38-9.87) 0.026 0.94 (0.06-1.85) 0.961 0,51 (0,03-2,27) 0,651

Ischemic heart disease 3,62 (1,48-8,85) 0,005 0,21 (0,05-0,78) 0,06 - -

*Adjusted IHD

This is a potentially valuable work, although the reviewer is not certain how many surgeons (internationally) perform single anesthetic bilateral TKA on patients who are morbidly obese. The authors have certainly demonstrated how this results in a substantial number of patients who experience complications (compared with non-morbidly obese patients). But, how will this study impact practice, if most surgeons don't perform SBTKA in this patient population?

The reviewer still wonders if the authors could provide an "upper limit" for safe SBTKA. Is it really 39.9 kg/m2?

RESPONSE: In this study, it was not aimed to determine an upper limit for safe SBTKA. This may be the subject of a different study. In our study, we aimed to analyze the safety of SBTKA in MO patients.

Thank you for your interest in this subject matter, for reviewing your experience, and for sharing this information. Best wishes as you continue to refine your work.

Reviewer #2: In this article, Bedrettin Akar presents the results of his comparative study in patients with simultaneous bilateral TKA and obesity. The patient groups differ primarily in their BMI, below and above a BMI value of 40 kg/m2. The author was able to include a total of 494 patients in the study, with a large difference in group size (65 vs. 429 patients).

Statistical analysis of the retrospective data was performed about wound healing problems (WHP), mobilization time (MT), operation time (OT), prosthesis infections, aseptic loosening (AL), early complications, revisions, and length of hospital stay. As expected, patients in the group with BMI >40 showed significantly more complications and worse clinical outcomes of arthroplasty. Therefore, the author recommends that patients with BMI >40 kg/m2 should be encouraged to reduce their weight. If this is not possible, SBKA should not be performed and the patient should be aggressively educated about the limitations, complications, and dangers.

The article contributes to the care of obese patients, whose numbers are steadily increasing and pose a increasing challenge to healthcare systems.

Therefore, we consider the article worthy of publication.

However, we again recommend a critical review. For example, the author should explain abbreviations when they are first used (see Introduction: MO). In addition, the differences between the groups would be even clearer if a higher degree of matching had been done preoperatively (age, sex). Here, Scandinavian studies on peri-implant issues show significant differences, but these studies are based on the analysis of large register cohorts, which were not available to the author. Perhaps he can briefly explain here why, in his opinion, it was not possible matching the groups even better.

RESPONSE: At the beginning of our study, we considered the patient differences between the two groups. We randomized our patients and reduced the obese group to 138 (67.32%) patients. However, we observed that the results of the two different groups did not change and we decided to leave the obese group as 429 patients in order to avoid patient loss in our study.

Table 1: Statistical Analysis

 Obese Group n:138 Morbidly obese Group n:65 p

Age 78,50±5,63 67,51±7,28 0,0001*

Gender Male 13 9,42% 3 4,62% 0,236+

 Female 125 90,58% 62 95,38% 

Hospitalization Time (Days) 7,37±1,88 9,62±1,68 0,0001*

Operation Time (Minutes) 82,61±14,88 89,77±14,7 0,002*

Mobilization Time (Days) 1,43±0,56 1,98±0,63 0,0001*

Wound healing problems 4 2,90% 14 21,54% 0,0001+

Debridement 0 0,00% 4 6,15% 0,003ǂ

Medial Retinaculum Separation 4 2,90% 9 13,85% 0,003+

Long –term complications None 414 96,50% 58 89,23% -

 Aseptic Loosening 1 0,72% 4 6,15% 0,047ǂ

 Septic Loosening 2 1,45% 1 1,54% 0,946ǂ

 Perprostatic Femur Fracture 4 2,90% 2 3,08% 0,925ǂ

Short –term complications None 125 90,58% 51 78,46% -

 AKI 5 90,58% 10 15,38% 0,013+

 DVT 8 90,58% 2 3,08% 0,691ǂ

 Pulmonary Embolism 0 90,58% 2 3,08% 0,049ǂ

Diabetes Mellitus 29 21,01% 13 20,00% 0,868+

Hypertension 77 55,80% 36 55,38% 0,956+

Ischemic Heart Disease 5 3,62% 8 12,31% 0,04+

Renal Disease 12 8,70% 5 7,69% 0,810+

Pulmonary Disease 5 3,62% 4 6,15% 0,414+

Venous insufficiency- lymphedema 5 3,62% 3 4,62% 0,735+

Thyroid Disease 2 1,45% 2 3,08% 0,436+

*Independent t test + Chi-square test ǂFisher's Reality Test

Reviewer #3: Bedrettin et al, aimed to analyze the safety of simultaneous bilateral total knee arthroplasty (SBTKA) surgery in morbidly obese patients compared to obese patients. The study found that SBTKA surgery is not feasible in morbidly obese patients due to the high complication rate and unsatisfactory clinical outcomes. The authors suggest that staged surgery be preferred and that the patient's request to undergo SBTKA should not be taken into account. The study used appropriate statistical analysis methodology, including univariate and multivariate logistic regression analyses, descriptive statistical methods, and tests for normality and comparison of variables between groups. However, the study has some limitations, such as the retrospective nature of the study which was addressed shortly by the authors and the use of one surgeon for all cases which has to be discussed as a limitation. The main findings of the study were adequately discussed in the Results and Discussion sections of the paper. Overall, the study provides valuable insights into the safety of SBTKA surgery in morbidly obese patients and highlights the need for further research on the treatment of morbidly obese patients.

Minor issues are the required a detailed check to enhance its readability, consistent writing styles between British and American English has to be corrected (ex. Analyzes, analyses). Also it is confusing have one group abbreviated (MO and the other with full name) I recommend using MO group and O group throughout the manuscript. 

However, two major issues shall be addressed:

1- The introduction: The authors briefly mention the high prevalence of obesity and the increased risk of complications associated with TKA surgery in obese patients. However, the introduction does not provide a detailed discussion of the critical need for this study in the field. The authors do not provide a comprehensive review of the existing literature on the safety of simultaneous bilateral TKA surgery in morbidly obese patients, nor do they discuss the limitations of previous studies. Therefore, while the introduction provides some motivation for the study, it should have been more comprehensive in its discussion of the critical need (gap of knowledge) for this study in the field. Furthermore, the frequency of bilateral operation in these cases should be introduced locally and internationally.

RESPONSE: The names of the groups have been corrected.

The following has been added to the manuscript:Considering that 40% of the world population is obese, this clinical problem continues with increasing incidence [1]. Obesity is a preventable-modifiable risk factor. Studies examining the trends in obesity have shown that its prevalence has risen in all age groups, regardless of ethnicity or socioeconomic status [1,2]. From 1999 to 2018, the prevalence of obesity increased from 30.5% to 42.4% in the United States. Notably, the obesity prevalence was 40% among adults aged 20 to 39 years, 45% among adults aged 40 to 59 years, and 43% among the group aged 60 years and older. Obesity is a strong risk factor for osteoarthritis of the knees [3]. The incidence of total knee arthroplasty (TKA) is high in obese patients. However, the frequency of simultaneous bilateral TKA varies in the literature is low, between 2 and 7% of all TKA.

2- the fact that all the patients were operated by the same surgeon and in one center is a limitation of the study. This is because the results of the study may not be generalizable to other centers or surgeons, as different surgeons may have varying levels of experience and skill, and different centers may have different resources and protocols for managing surgical procedures. Additionally, having a single surgeon and center involved in the study may introduce bias into the results. For example, the surgeon may have a preferred technique or approach that could influence the outcomes of the surgery, and the center may have different patient populations or healthcare resources that could impact the results.

Therefore, it is important to interpret the study's findings within the context of this limitation, and to consider the need for further research in other centers and with different surgeons to confirm the study's results and to better understand the safety and effectiveness of simultaneous bilateral knee arthroplasty in morbidly obese patients.

RESPONSE: You are absolutely right. This sentence was removed from the article and included in the limitations as follows:“Furthermore, the fact that all patients were operated by the samesurgeon and in a single center was a limitation of the study.”

---

## [Decision Letter · Decision Letter 1]

30 May 2023

IS THERE AN INDICATION FOR SIMULTANEOUS BILATERAL KNEE ARTHROPLASTY IN MORBIDLY OBESE PATIENTS? SHOULD THE PATIENTS’ REQUEST FOR SIMULTANEOUS OPERATION BE CONSIDERED?

PONE-D-22-20821R1

Dear Dr. Akar,

We’re pleased to inform you that your manuscript has been judged scientifically suitable for publication and will be formally accepted for publication once it meets all outstanding technical requirements.

Kind regards,

Sameh Attia, MS

Academic Editor

PLOS ONE

Additional Editor Comments (optional):

Reviewers' comments:

Reviewer's Responses to Questions

**Comments to the Author**

1. If the authors have adequately addressed your comments raised in a previous round of review and you feel that this manuscript is now acceptable for publication, you may indicate that here to bypass the “Comments to the Author” section, enter your conflict of interest statement in the “Confidential to Editor” section, and submit your "Accept" recommendation.

Reviewer #2: All comments have been addressed

2. Is the manuscript technically sound, and do the data support the conclusions?

Reviewer #2: Yes

3. Has the statistical analysis been performed appropriately and rigorously? 

Reviewer #2: Yes

4. Have the authors made all data underlying the findings in their manuscript fully available?

Reviewer #2: Yes

5. Is the manuscript presented in an intelligible fashion and written in standard English?

Reviewer #2: Yes

6. Review Comments to the Author

Reviewer #2: The author has taken all the reviewers' comments and revised them. I recommend acceptance of the manuscript for publication.

7. PLOS authors have the option to publish the peer review history of their article (what does this mean?). If published, this will include your full peer review and any attached files.

Reviewer #2: No

---

## [Editor Report · Acceptance letter]

1 Jun 2023

PONE-D-22-20821R1 

IS THERE AN INDICATION FOR SIMULTANEOUS BILATERAL KNEE ARTHROPLASTY IN MORBIDLY OBESE PATIENTS? SHOULD THE PATIENTS’ REQUEST FOR SIMULTANEOUS OPERATION BE CONSIDERED? 

Dear Dr. Akar:

I'm pleased to inform you that your manuscript has been deemed suitable for publication in PLOS ONE. Congratulations! Your manuscript is now with our production department. 

Kind regards, 

on behalf of

Dr. Sameh Attia 

Academic Editor

PLOS ONE